# Colorimetric, Naked-Eye Detection of Lysozyme in Human Urine with Gold Nanoparticles

**DOI:** 10.3390/nano11030612

**Published:** 2021-03-01

**Authors:** Paula M. Castillo, Francisco J. Fernández-Acejo, Jose M. Carnerero, Rafael Prado-Gotor, Aila Jimenez-Ruiz

**Affiliations:** Department of Physical Chemistry, University of Seville, 41012 Seville, Spain; pcastillo@us.es (P.M.C.); franfdez24@gmail.com (F.J.F.-A.); jcarnerero2@us.es (J.M.C.)

**Keywords:** gold nanoparticles, lysozyme, urine, colorimetry, CIELab, naked-eye detection, lysozymuria, proteinuria

## Abstract

The stabilizing effect of lysozymes to salt addition over a gold colloid are exploited in order to detect lysozymes in human urine samples. The present research is aimed at the development of a fast, naked-eye detection test for urinary lysozymuria, in which direct comparison with a colorimetric reference, allows for the immediate determination of positive/negative cases. CIEL*a*b* parameters were obtained from sample absorbance measurements, and their color difference with respect to a fixed reference point was measured by calculating the ΔE_76_ parameter, which is a measure of how well the colors can be distinguished by an untrained observer. Results show that a simple and quick test can reliably, in less than 15 min, give a positive colorimetric response in the naked eye for concentrations of a urinary lysozyme over 57.2 µg/mL. This concentration is well within the limits of that observed for leukemia-associated lysozymurias, among other disorders.

## 1. Introduction

Lysozyme, which is a cationic protein at physiological pH, can be eliminated through the urine because of its low molecular weight. Its charge makes it stand out among most proteins in the urine (such as albumin and globulins), which have a negative or virtually neutral charge at a biological pH [1,2]. However, as a general rule, proteins are only found in trace amounts in human urine samples obtained from healthy individuals. Some medical conditions involving either renal disfunction or the production of excess lysozyme can dramatically elevate lysozyme levels in urine (lysozymuria) [3,4]. Lysozymurias are associated with various renal disorders, such as hypokalemia, extrarenal infections, or a nephrotic syndrome [5], and have also been determined to be a distinct symptom of monocytic and myelomonocytic leukemia (both subtypes of acute myeloid leukemia) [3,4]. In the latter cases, lysozyme levels found in urine have been found to be abnormally high when compared to other disorders. Lysozymuria detection plays a key role in early diagnose of monocytic and myelomonocytic leukemias.

Noble metal nanoparticles have acquired great importance in the field of biomolecule detection due to their optical properties. The oscillation of the electronic cloud with the electric component of the light causes a strong band of absorption in the visible region, whose location (and, therefore, the color of the sample) depends on the colloid intrinsic properties and its environment [6]. In this way, when these nanoparticles interact with the biomolecule to be detected, a change of color takes place in the solution [7,8]. Specifically, gold nanoparticles (AuNPs) have been extensively used to the detection of a varied and wide variety of compounds, from proteins to DNA [9,10,11,12,13]. The procedure is quite simple. The interaction between the analyte and the AuNPs induces approximation (aggregation) of the nanoparticles, leading to nanoparticle plasmon coupling and a change of color from red to blue in the solution. Reverse procedures are also possible with analytes being employed to either prevent aggregation, or further the distance between aggregated particles, causing a blue-to-red color shift [14]. The former is the case for lysozymes, as shown in Scheme 1.

Along those guidelines, multiple strategies have been carried out, from the simplest to very complex ones, in order to detect a myriad of proteins and other biological molecules. For example, cholesterol has been detected with AuNPs functionalized with cholesterol oxidase, interacting with the analyte, who makes junction points between the particles and causes the previously mentioned chromatic changes (inducing shifts of the absorbance plasmon band of c.a. 35 nm) [15]. Lin and coworkers have designed a system where the presence of the protein target (highly effective to VEGF, vascular endothelial growth factor) catalyze the formation of double chain DNA dendrimers, which cannot protect AuNPs of the aggregation induced by salt addition [16]. A similar procedure was employed by He and colleagues, who detected the AFP (Human α-fetoprotein) using the rolling circle amplification reaction as a cornerstone. The presence of the target protein generates oligonucleotides that do not prevent the aggregation caused by chloride sodium [13]. Proteins can interact directly with the nanoparticle surface, even accumulating in several layers and causing (small or big) changes in the absorption band [17].

A good number of studies have developed protocols for the detection of a lysozyme using gold nanoparticles with some of them in biological media [18,19,20]. Wang et al. carried out urinary lysozyme determination by employing gold nanoparticles in a really notable work that, nevertheless, employs resonance light scattering spectra for detection. Wang et al. also use lysozyme as a direct nanoparticle aggregator, so their positive response corresponds to a blue tint while a negative response gives a red tint. This method gives a lower detection threshold as lysozyme acts as an aggregating agent even when present in very small quantities [21], but it opens the possibility for samples to turn back to red when high quantities of lysozyme are present, as lysozyme presents a marked protective effect over nanoparticle aggregation [21]. Detection by aggregation is also highly vulnerable to any other interferents that may cause particles to agglomerate, such as cationic compounds. They do compensate for this by doing a very comprehensive work on pre-processing urine samples, including a protocol for the elimination of human serum albumin (HSA). This is a protein interferent that may appear alongside lysozyme in renal-damage-induced proteinuria [22]. However, their work is carried out by employing a complex detection technique that, while not as vulnerable as naked eye testing for all those factors, is also expensive and may not be available at all at testing locations.

Fei Fu et al. [19] do detect lysozyme in human blood samples by using plasmon resonance light-scattering of gold nanoparticles, but they need to consider the use of a peptidoglycan to bind to lysozyme, and they base their work on a luminescence response, involving complex measurements and equipment. Lihua Lu et al. [23] employ a novel Ir (III) complex to generate a strong luminescence response by means of a duplex DNA with a TBAG-quadruplex tail for the detection of lysozyme. The protein induces duplex dissociation of a complex of Ir (III) to generate the luminescence response.

Other authors, such as Jing Luen Wai and Siu Yee New [20], have used non-citrate AuNPs, specifically cysteamine-stabilised AuNPs (cysAuNPs). The great advantage is that these nanoparticles can directly interact with DNA with an anionic charge, without the need for an inert electrolyte. However, in addition to functionalizing gold, they work with lysozyme-binding aptamer (LBA) and their study only refers to aqueous media. There are other, very recent studies to detect lysozyme, but all of them either employ aptamers or work with non-biological samples, such as wine samples [24,25]. Other authors use nanorods [26] or carbon nanotubes, which are also functionalized with aptamers [27]. Lixiang Zuo does lysozyme detection in urine samples, but their method involves preparation of Mn-doped ZnS quantum dots [28] as does the one employed by Zhenli Qiu and coworkers for spiked serum samples [29].

Since the aim of our work is to develop a cheap, easy to use, naked-eye lysozyme sensor that can be employed without the need for expensive equipment or training, we have employed CIEL*a*b* colorimetric parameters for the determination of the quality of our results. The CIEL*a*b* colorimetric system is an absolute color coordinate system that is based on the theory of opposite colors developed by Schrödinger [30]. That is, red and green cannot be perceived at the same time, and neither can blue and yellow due to them generating eye responses that are opposite of each other [31]. The system is defined around an illuminant (which emulates external light under controlled circumstances) and an observer (which emulates human eye structure) function [32]. With those two conditions, a white color point is generated as the zero, and color intensities are mapped alongside three axes: L*, which corresponds to the luminance of the system, a*, which represents red against green tint, and b*, which does the same to yellow versus blue (Scheme 2).

CIEL*a*b* color coordinates do not depend on the device employed for reproduction (as do other systems, such as RGB for digital color or CMYK for printing), but, instead, univocally and universally define a specific tone. They are also useful for the determination of color differences, as the distance between two color points (ΔE) expressed in the L*a*b* reference system determine how different they are, or, in other words, if they will be perceived as different tones by an observer. The minimum distance between two different colors and tones of the same color is known, and has been extensively studied [33]. A value of the ΔE parameter equal to or over 2.3 for two colorimetric points is known as the Just Noticeable Difference (JND) and is considered the minimum distance needed for those two points to be considered distinguishable, under good conditions, by a human observer at a naked eye.

There is precedent for the use of CIEL*a*b colorimetric parameters for the analysis of the interaction of gold nanoparticles with a ligand [34]. However, the present work makes use of CIEL*a*b* parameters and their difference to determine if naked-eye test results will be read as different (positive response) or not (negative response) by a human observer during a field test, but the use of a colorimeter is not a requisite to carry out the testing. In this field, CIEL*a*b* parameter quantification has most notably been employed such as by Mbambo et al. in 2019 [35] who used color reproduction to create a digital color scale for the development of a salinity test for saline and estuarine water. Similar tests employing RGB colors for digital simulation on a smartphone have also been developed recently for the detection of sibutramine on food products [36].

In this work, we present a direct, simple, and novel method that allows for naked-eye detection of micromolar range concentrations of lysozyme in urine. To this end, non-functionalized, anionic gold nanoparticles have been used in order to exploit both the previously mentioned strong positive charge of lysozyme and the extraordinary optical properties of colloidal gold. The strong blue color of aggregated gold nanoparticles has been taken as the reference point. NaCl has been used as the agglomerating agent, and sodium citrate has been added after synthesis in order to act as a stabilizer, helping to reach lower detection limits. Nanoparticle size, concentration of all reactants involved, and addition order have been carefully optimized in order to develop a stable, solid, and reliable system with clearly distinguishable positive (in the presence of lysozyme) and negative signals.

## 2. Materials and Methods

### 2.1. Nanoparticle Synthesis and Stabilization

Spherical citrate-capped gold nanoparticles were synthesized by a modified Turkevich method, involving direct reduction of HAuCl_4_ salts (Sigma-Aldrich, Darmstadt, Germany, ref. number 520918) with sodium citrate (Riedel-de Haën, Honeywell International, Charlotte, NC, USA, ref. number 32320) at 95 °C with magnetical stirring [37]. The resulting synthesis was characterized by transmission electron microscopy (TEM) and the images obtained were analyzed using ImageJ software. For preliminary tests, mean particle diameter was found to be 14.1 ± 0.9 nm, with a circularity over 92% for all cases, and the final nanoparticle concentration in the synthesis was estimated at 4.1 × 10^−9^ M. Working tests were carried out with a synthesis with a mean particle diameter of 15.0 ± 1.4 nm, and a concentration of 3.6 × 10^−9^ M. In both cases, nanoparticle synthesis was found to be monodisperse (less than a 15% size dispersion; see Figure 1).

Preliminary experiments with urine samples showed that the post-synthesis addition of sodium citrate greatly increased nanoparticle stability and allowed for a clearer colorimetric signal and a lower detection limit. The optimum final citrate concentration was found to be 2.3 × 10^−3^ M. Higher concentrations of added sodium citrate were found to have the opposite effect due to the influence of the increase in the medium ionic strength being greater than that of NP stabilization by citrate adsorption. On the other hand, when citrate concentration was increased during the synthetic procedure, nanoparticle precipitation was observed. For this reason, an additional step involving sodium citrate addition was incorporated into all sample preparation protocols, as described below.

### 2.2. Preliminary Testing

#### 2.2.1. Sample Collection and Processing for Preliminary Testing

Urine samples from healthy subjects (labelled 1 to 3) were collected and known concentrations of lysozyme were added to each sample. Concentrations of lysozyme in the as-prepared urine solutions accounted for urinary lysozyme concentrations ranging from 10^−5^ M (143.1 µg/mL) to 5 × 10^−8^ M (0.72 µg/mL) and a zero sample (urine, no lysozyme). A 50% predilution (1 mL sample + 1 mL water) was then carried out in order to minimize urine color interference.

#### 2.2.2. Preliminary Testing Protocols

Colorimetric analysis was carried out by mixing, in this order, 200 µL AuNPs + 900 µL deionized water + 200 µL sodium citrate + 200 µL prediluted urine samples + 500 µL NaCl. Final concentrations were [AuNPs] = 3.7 × 10^−10^ M, [Citrate] = 2.3 × 10^−3^ M, [NaCl] = 0.05 M. Working solutions were completely stirred and left to react for 5 min for color stabilization. In addition to the test solutions, a sample was prepared containing no added lysozyme (AuNPs + water + citrate + urine + NaCl, zero sample) and another without urine or lysozyme (AuNPs + water + citrate + NaCl, blank sample). The reproducibility of the method was evaluated by measuring three separate solutions prepared from the same initial sample (labeled as samples 1.1 to 1.3).

### 2.3. Working Protocol

#### 2.3.1. Urine Collection and Processing

Ten urine samples from healthy subjects (labelled A to J) were collected and known concentrations of lysozyme were added to each sample. Preliminary test results allowed for the first approximation to the positive response threshold, and, therefore, lysozyme concentrations were updated. Some concentration points that were observed to be too low for detection were removed, and more points were added in the turning zone in order to better pinpoint the detection threshold. Concentrations of lysozyme in the as-prepared urine solution now accounted for urinary lysozyme concentrations ranging from 143.1 µg/mL to 7.15 µg/mL, plus a zero sample with no added lysozyme.

As per the preliminary test results, urinary salt concentration was considered to be a possible source of interference in which one where the proposed 50% urine dilution employed during preliminary sample preparation failed to properly address. Pre-treatment protocols were then updated as follows. The lysozyme was added to native urine samples in a first step 1:2 dilution. Then, the refraction index of all spiked urine samples and that of water was measured at room temperature (between 22.8 and 23.5 °C) in an Abbe WYA-1S refractometer, and a ratio *r_i_* = *r_sample_*/*r_water_* was then calculated. Deionized Milli-Q water was added to the samples until *r_i_* < 1.002. Those diluted urine samples were then employed for sample preparation.

#### 2.3.2. Sample Preparation

In light of the preliminary tests’ results, a few modifications were made in order to develop the final working protocol. In this order, 400 µL AuNPs (1.9 × 10^−8^ M) + 500 µL deionized water + 400 µL sodium citrate (1.16 × 10^−3^ M) + 200 µL prediluted urine samples were mixed. A blank containing no urine, which was replaced by the equivalent volume of deionized water, was also prepared in the same way for each sample. A 1 M NaCl solution was then added drop-by-drop, while stirring, to this urine-free blank until a blue color shift was observed. The same volume of NaCl was then added to the sample preparations. This step allowed us to pinpoint the exact quantity of salt needed to change the color of a given gold preparation, eliminating the possibility of a false positive caused by a deficit of NaCl.

### 2.4. Sample Measurement and Obtainment of CIEL*a*b* Parameters

For both the preliminary and the final working protocol, samples were completely stirred and left to react for 5 min for color stabilization before measurements were done. Colorimetric analysis was carried out by measuring transmittance in a Cary 500 UV-vis spectrophotometer (Agilent, Santa Clara, CA, USA) working at room temperature. XYZ color space measurements were derived from transmittance values according to the CIE (International Commission on Illumination) standards for a D65 illuminant and 2° standard observer [38]. In order to set up quantitative guidelines for the determination of a positive response color threshold, a mathematical conversion from XYZ to L*a*b* was carried out as described by the CIE [38].

Digital (RGB) simulation of the sample colors was also carried out. Conversion from the XYZ data to RGB was done in accordance with the equations found in https://www.easyrgb.com/en/math.php (accessed on 3 March 2014). Those colors are reproduced in this paper to graphically illustrate the results of our analysis, but were not employed for mathematical calculations.

The ΔE parameter, which evaluates a color difference between a sample and a reference color, was calculated from CIEL*a*b* parameters for all cases [39]. It is important to note that ΔE values were calculated in two different ways for the preliminary tests and for the final ones.

Preliminary tests: referenced to L*a*b* parameters of a blank sample, which does contain aggregated AuNPs without urine or lysozyme and, therefore, can be employed as a neutral reference point. However, in doing so, it was observed that zero-lysozyme samples could also show subtle color differences from the urine-free blank.Working tests: referenced to the mean L*a*b* parameters obtained for all zero (lysozyme-free) urine samples. In this way, a “neutral” reference that was not directly related to any of the samples was created, and residual matrix effects of urine that could induce color changes were accounted for and compensated.

For ΔE evaluation, the classical CIE76 formula (equivalent to the Euclidean distance between the reference and the sample color coordinates in a reference system) was used (Equation (1)).
(1)$$∆E_{76}=\;\sqrt{{\left(L^{*}-L^{*}{}_{Blank}\right)}^{2}+{\left(a^{*}-a^{*}{}_{Blank}\right)}^{2}+\;{\left(b^{*}-b^{*}{}_{Blank}\right)}^{2}}$$

## 3. Results

### 3.1. Preliminary Test Results

For the preliminary tests, upon NaCl addition, absorbance spectra were recorded, as shown in Figure 2. It is important to note that, for this representation, absorbance spectra were normalized in order to better show maxima position. Peak position showed a clear red shift (towards higher *λ*) at lower lysozyme concentrations, followed by blue shifting for concentrations of 71.5 µg/mL or over. The zero point of the series shows a smaller red shift than that observed for the lower lysozyme concentrations.

Naked-eye results (Figure 3) showed, in all but one of the cases, a clear red color (positive response) for urinary lysozyme concentrations of 71.5 µg/mL or higher after NaCl addition. Zero samples were blue in all cases, as were the blanks (not shown). No blue color was found in any of the samples prior to salt addition. All samples showed a color change when the lysozyme concentration was over the previously mentioned detection threshold, even though, in the outlier case, a purple tint instead of red was reached as the final point.

In order to set up quantitative guidelines for the determination of a positive response color threshold, transmittance measurements were done and CIELab parameters were derived from the results. The obtained L*a*b* values were then used for digital simulation of the sample colors, in order to allow for a clearer color observation without external illumination interference (Figure 4). Three freshly-collected samples, 1, 2, and 3 (outlier case), were analyzed, and the reproducibility of the method was evaluated by measuring three separate solutions prepared from the same initial sample (labeled as samples 1.1 to 1.3). From those measurements, standard deviation of the ΔE parameter was found to be under 10% for all cases.

In this case, ΔE values were measured in reference to the L*a*b* parameters of the blank sample, which contains aggregated AuNPs without urine or lysozyme. For ΔE evaluation, the CIE76 formula was used (see Equation (1)). Results were contrasted against those obtained from the more complex, corrected ΔE CIE94 formula [40].
(2)$$∆E_{94}=\;\sqrt{\frac{\Delta L^{*2}}{k_{L}S_{L}}+\frac{\Delta C^{*2}}{k_{C}S_{C}}+\;\frac{\Delta H^{*2}}{k_{H}S_{H}}}$$
(3)$$C^{*}=\;\sqrt{a^{*2}+b^{*2}}$$
(4)$$\Delta C^{*}=C^{*}-C^{*}{}_{Blank}$$
(5)$$\Delta H^{*}=\;\sqrt{\Delta a^{*2}+∆b^{*2}-∆C^{*2}}$$
(6)$$S_{L}=1$$
(7)$$S_{C}=1+0.045\;C^{*}{}_{Blank}$$
(8)$$S_{H}=1+0.015\;C^{*}{}_{Blank}$$
(9)$$k_{L\;}=\;k_{S}=\;k_{H}=1$$

Despite the CIE94 system being more accurate at addressing smaller color differences [39] and non-uniform color perception [31] than the CIE76, less than a 2% difference was found in all cases between CIE76 and CIE94 values (Table 1). Since the precision gain was deemed not enough to justify the use of a more complex system, the CIE76 system was employed through the rest of our study.

As noted before, a ΔE value of 2.3 or over is termed the JND (Just Noticeable Difference) threshold for close tones to be considered as distinguishable by an untrained observer [32]. Two tones that present a ΔE color difference under the JND are harder or impossible to distinguish from each other, while two tones whose ΔE is over the JND will be interpreted as different. The JND threshold is indicated by a dashed line in Figure 5. The most significant finding of this series is that the color difference value ΔE between the blank and zero samples in normal cases can be found in the ΔE = 2.3 to 2.6 bracket. This means that zero samples (lysozyme-free urine samples) can be distinguished from the blank sample by direct comparison, thereby, posing a risk for false positive results.

### 3.2. Working Test Results

A working test protocol differed from the preliminary one in some ways, as we tuned up our work conditions to enhance results. For a start, after the 1:2 predilution employed for native urine samples, their refractive index was also measured prior to starting the test. Pre-diluted samples were then diluted again until their *r_i_* = *r*_sample_/*r*_water_ was under 1.002 in order to compensate for their different salt content, and then the rest of the protocol was followed.

Nanoparticle volume in preparations was raised to 400 µL from the original 200 µL, aiming for the obtention of a more intense tint without compromising detection thresholds. In addition, the “fixed volume” preparations employed through preliminary tests were changed to “dynamic volume” ones. The final volume in the detection cuvettes was not always the same, but was compensated to be as small as possible while still having the appropriate NaCl concentration to induce nanoparticle aggregation in the absence of lysozyme. To this end, each urine series was accompanied by a control sample, in which both lysozyme and urine volume were replaced by deionized water. This control sample was then added NaCl 1 M drop-by-drop until the blue tint appeared, and then the same NaCl volume was added to each point in a given series. This approach has two benefits: first, as AuNPs synthesis tend to slightly differ in concentration between batches, the minimum NaCl volume needed for aggregation may not always be the same, and, second, the risk of a “false positive” caused by a deficiency of added NaCl is also averted. Minimizing preparation volume also has the added benefit of avoiding excess AuNPs dilution, therefore, allowing for clearer red and blue tints to be observed.

As can be seen in Figure 6, the red shift of the absorbance maxima upon NaCl addition can now clearly be observed for samples 0 to 28.6 µg/mL, while upper concentrations retain the red color of unaggregated nanoparticles and appear blue-shifted in relation to the zero. No “bouncing” of the zero-lysozyme sample is observed.

Figure 7 shows the RGB simulation of naked-eye test results. Again, for all samples, a clear red tint appears at higher lysozyme concentrations. For some of the samples, a positive response can be observed for concentrations as low as 28.6 µg/mL. It is also interesting to note that the reddish effect observed on some preliminary samples in the absence of lysozyme (0 µg/mL, see Figure 4) was corrected by the changes made to the improved protocol, minimizing the risk of a false positive.

For the experimental L*a*b* parameters of the 10 samples, ΔE values were derived. In this case, however, the reference point from which the color difference was calculated was the numerical mean of the three parameters for the zero sample of all series. In this way, the calculus compensated for the small red shift that had previously been observed for some of the zero samples. It is important to note that there was a little difference in the results when each ΔE was calculated by taking its own series zero as a reference. However, a common reference colorimetric point that can, for example, be reproduced digitally or in printing is more useful in a real setting where there is no access to a known lysozyme-free sample. The resulting L*a*b* values taken as reference were L* = 94.4, a* = −0.62, b* = −2.7.

As can be observed in Table 2 and Figure 8, for a lysozyme concentration of 114.4 µg/mL or over, all 10 samples gave positive results that could be distinguished from the reference at a naked eye. Moreover, nine out of ten cases were also over the JND threshold for lysozyme concentrations as low as 57.2 µg/mL, and four of them also gave a positive response when lysozyme concentration was halved. For the zero samples, none of them deviated so much from the reference as to constitute a false positive. This represents a marked improvement from the preliminary results in which zero samples could appear over the JND threshold.

## 4. Discussion

### 4.1. Preliminary Test Results

Absorbance measurements of the lysozyme-spiked urine samples (see Figure 2) clearly showed the protective effect of lysozyme over gold nanoparticle aggregation for the more concentrated samples, which are 71.5 µg/mL and over. On the other hand, in light of those results, it becomes apparent that the multitude of processes involved with analyzing biological samples do cause a clear widening of the peaks, which is more apparent for blue-shifted results (14.3 µg/mL and under). This widening causes the exact position of the maximum to become ambiguous. This method of determining a positive response also requires spectrophotometric measurements.

For naked-eye tests, the positive signal threshold was considered to correspond to values over the JND (ΔE values of 2.3 and over), which is understood to be the smallest ΔE value needed for two tones to be perceived as different by untrained observers. As seen in Figure 4 and Table 1, results corresponding to urinary lysozyme concentrations of 71.5 µg/mL and over (clear red tint in the solution at naked eye) did reflect in all cases, but the outlier, on values of ΔE over 5, which are generally accepted to correspond to a clear color difference between two samples [33,41]. The color difference between concentrations of 71.5 and 143.1 µg/mL was almost non-existent. If all positive responses are analyzed globally, the positive response ΔE bracket extends from 5.2 to 5.7 with a mean value of ΔE = 5.5. Therefore, in the majority of the cases, the color difference observed after tuning up the system (15-nm anionic, citrate-capped AuNPs under optimum citrate and NaCl concentrations) allows for a clear distinction between a lysozyme-containing and a lysozyme-free urine sample. By direct comparison with a freshly prepared blank sample, naked-eye lysozyme detection in urine can, therefore, be carried out in a quick and easy way.

However, although the positive-blank color difference is enough to be noticeable at a glance in the outlier case 3, the difference between the positive response of this sample 3 at [lys] > 71.5 mg/mL (ΔE_76_ ≈ 2.8), and the 1 and 2 control samples with [lys] = 0 (ΔE_76_ ≈ 2.4) is not big enough to unequivocally ascertain the presence of lysozyme by the naked eye observation without either risking false positives for 1 and 2 or a false negative for 3.

The presence of a purple tint on zero-lysozyme samples can be attributed to the presence of trace amounts of other proteins. Those data are in accordance with Wang et al. [18] who reported similar observations when analyzing lysozyme-spiked urine samples through the Plasmon Resonance Light-Scattering (PRLS) technique. The higher grade of AuNPs aggregation found in sample 3, which leads to a greater blue color intensity, might be due to an abnormally high salt content in the initial sample.

### 4.2. Working Test Results

With the modified protocol, the normalized absorbance spectra of the samples (Figure 6) shows a similar behaviour to that observed for the preliminary test results. Lower lysozyme concentration samples do experience a red shifting of the absorbance spectra due to nanoparticle aggregation, while higher concentration samples remain on the green-blue absorbance zone. In this case, although the general tendency is clear, peak widening becomes even more apparent than it was during the preliminary phase of the study.

As for naked-eye testing, changes made to the preliminary testing protocol proved useful in pinpointing the detection threshold of the method. The use of a “real” reference point for the test series, obtained from the mathematical mean of the three L*a*b* parameters for the zero-lysozyme samples, meant that the possible purple tint that may appear in a lysozyme-free sample is accounted and compensated for. The reference shift also meant that less lysozyme-containing samples tested over ΔE = 5, that is, they were less distinguishable from the real reference than they were from the AuNPs + NaCl reference employed for the preliminary tests. However, all positives still tested over the JND, meaning that the color difference is still enough for the system to work.

On this ten-sample series, no false positives were observed. That is, all zero samples tested well under ΔE = 2.3 and cannot be considered as distinguishable from the reference at a naked eye. On the other hand, nine out of the ten samples showed a color change over the JND when lysozyme concentrations over 57.2 µg/mL were employed, reading as positives. For the outlier case, the color change appeared at a higher concentration threshold of 114.4 µg/mL. Concentrations under that threshold can be considered as “false negatives” in which the color test reads as a negative even in the presence of measurable lysozyme concentrations. More importantly, four out of ten samples also showed color changes over the JND for concentrations as low as 28.6 µg/mL, therefore, also testing as positives.

Based on that data, and on mean ΔE values and error margins that can be found in Figure 8, we propose a detection limit of 57.2 µg/mL for our naked-eye urinary lysozyme test.

### 4.3. Result Evaluation

Finally, in order to evaluate the usefulness of the proposed method, the obtained lower threshold limit needs to be put in context. Urinary lysozymuria associated with monocitic and myelomonocitic leukemia was found in a classic work to range between 25 and 420 µg/mL of lysozyme [3]. A range of 62 to 211 µg/mL has been observed for acute monocitic leukemia, and a range of 0 to 87 µg/mL has been observed for acute myelomonocitic leukemia [42]. It is important to note that urinary lysozymuria is virtually absent in other leukemia types, so its early detection may help narrow leukemia type diagnoses in a clinical setting. Lysozymuria has also been found associated with varied renal diseases, up to approximately 30 µg/mL [5]. Patients suffering from diabetic nephropathy can also show increased urinary lysozyme levels, up to 10 µg/mL [43].

As for interference testing, human albumin (HSA), which is a protein commonly found in urine [1,44], has been found to interact with gold nanoparticles [45,46]. Globulins, which is another common urinary protein family [47], have also been reported to have the same effect [45]. Those interactions, where the adsorption of proteins over the particle leads to the formation of a protein corona, may cause a small degree of protection in the presence of salt, leading to a purple color. In concrete, HSA may appear alongside or instead of lysozyme in some kinds of proteinurias, such as those induced by renal damage or diabetes [48]. Protocols for elimination of those proteins by precipitation have been proposed by Wang et al. [18]. Since the isoelectric point of those proteins is below 7 in all cases [1], they remain uncharged at a biological pH, while lysozyme presents a cationic charge. This means that selective elimination of contaminant proteins, if suspected, can be carried out relatively easily.

## 5. Conclusions

As stated before, a good number of studies have dealt with lysozyme detection in the last few years. However, almost all of the existing studies employed complex detection techniques that may or may not be available in a clinical setting, or made use of expensive nano-systems (for example, aptamer-based tests) in which production costs may be a limiting factor when trying to develop a first-approach commercial test to lysozyme and urinary proteinuria. Most of the existing literature also deals with methods developed either in aqueous media or non-biological samples, such as wine samples, with urine examples being scarce even in the face of the preeminence of lysozymurias and other proteinurias associated with various disorders.

In light of the data exposed throughout this manuscript, it becomes apparent that our method is able to detect urinary lysozyme concentrations within the range distinctly associated with both monocitic and myelomonocitic leukemia, among other pathologies. Both lower production costs and shorter detection times in relation to existing tests point to a method that could become a helpful aid in early detection of those pathologies, which is crucial to save lives. It is also a fast (under 10 min), easy, and inexpensive system to do first-line testing for lysozymuria, requires no specialized equipment to be carried out, and can be read by a simple, naked-eye color assessment with a reference.

## Data Availability

The data presented in this study are available on request from the corresponding author.

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
