# Peer review of "Colorimetric, Naked-Eye Detection of Lysozyme in Human Urine with Gold Nanoparticles"

_nanomaterials, 2021, doi:10.3390/nano11030612_

Round 1

Reviewer 1 Report

The paper “Colorimetric, naked-eye detection of lysozyme in human urine with gold nanoparticles” by Castillo et al. deals with the possibility of easy and fast detection of urinary lysozyme achieved by colorimetric approach, employing gold nanoparticles as colorimetric agent.

The manuscript has been substantially improved and enriched in the experimental and results sections. Now it sounds well organized and the results more convincing.

Therefore, the paper is suitable for publication in Nanomaterials, after a mandatory carefull spell checking.

In the following, a list of mispelling and typos to be corrected.

l.13 determinarion -> determination

l.37 their optic properties -> optical

l.39 colloid’s intrinsic -> colloid

  1. 56 efective to VEGF -> effective

l.57 the agregation induced -> aggregation

l.58 AFP (Human α-feroprotein -> fetoprotein

l.61 nanoparticles’ -> nanoparticle

l.80 PGN ?

  1. 83-84 The protein is responsible for the duplex dissociation using after the complex of Ir (III) to generate the luminescence response. ???? -> it is obscure.
  2. 110 an -> a

l.128 e l. 251 afforementioned -> aforementioned

l.155 medium’s -> medium

l.207 obtention? -> obtainment

l.210 uV -> UV

in Fig.2 trace relative to the 0.71 sample is missing

l.272 the precission gain -> precision

l.292 the blanket ??? -> blank?

l.299 cubettes was -> cuvettes

l.303 drop-by-drop fashion -> drop-by-drop

l374 (ΔE76 -> 76 subscript

Author Response

We heartfully thank the reviewer for their methodical revision and comments, and we have carried out the corrections and spell-checking as required.

Reviewer 2 Report

Authors have addressed in a positive and exhaustive way all the concerns reported in the previous review.
So I suggest to accept the article.

Author Response

We thank the reviewer for their time and helpful commentaries.

This manuscript is a resubmission of an earlier submission. The following is a list of the peer review reports and author responses from that submission.

Round 1

Reviewer 1 Report

Ruiz et al. have reported a paper concerning the colorimetric detection of Lysozyme in Human urine with gold nanoparticles.

The paper is quite well written and the idea to detect by naked-eye a colorimetric change may be quite interesting. However, at this stage the paper is not enough described to be published. I have some concerns, as follow:

  1. The references in the introduction are not sufficient to provide a sufficient background: I would suggest to describe the colorimetric properties of gold nanoparticles and the corresponding references. Also, I would add the effect of proteins on gold nanoparticles and what is already published in the literature concerning protein detection with gold nanoparticles.
  2. Characterization of the gold nanoparticles should be reported in the main text (TEM microscopy and UV-Visible measurements). I would suggest also to add the effect of NaCl on the nanoparticles absorption spectrum.
  3. I would suggest to add the UV-Visible absorption measurements, to make the colorimetric change clearer. Probably with these measurements the absorbance change may be followed in a better way, and spectral changes at different concentrations may be recorded (as example at 143.1 and 71.54 mg/ml).
  4. A little understanding of what is happening on gold nanoparticles by adding the salt and in presence of the lysozyme should be depicted with a scheme, in order to make also a non-expert readers understand the colorimetric change.
  5. At least three different measurements of each sample should be performed.
  6. When three measurements are performed, the mean value should be reported, and the error is the standard deviation of the mean. If there is a reason why the authors reported all the results in a table (Table 1), and not the mean values, this should be reported and emphasized.
  7. The absence of colorimetric change of a sample (sample C) should be described and rationalized. Why with this sample the system is not working?
  8. The authors wrote that also other urine proteins may have the same effect on the color of the solution. So what is happening if there are high concentrations of albumin or globulins in the urine? The effect of these interferers should be evaluated and measured.

Minor concerns:

pag.2, line 59: completely instead of completly

Reviewer 2 Report

In this manuscript, authors describe a colorimetric detection strategy for lysozyme in urine. The method should be soundly presented for naked-eye detection. However, there are many studies reporting colorimetric detection of lysozyme using gold nanoparticles (e.g., Chen et al., Langmuir, 24, 3654-3660, 2008; Huang et al., Analytical Methods, 4, 3874-3878, 2012; Li et al., Sensors and Actuators B, 147, 110-115, 2010 etc), and the present manuscript does not report new findings or significant advancement in the field. In addition, it is not clearly demonstrated whether any cross talk rather than lysozyme may interfere the colorimetric change of the gold nanoparticles. Overall, the manuscript should be substantially improved to be published in Nanomaterials. Particularly, overall writing can not clearly deliver the claim of this study. Also, authors should emphasize the importance of their study compared with others.

Reviewer 3 Report

The paper “Colorimetric, naked-eye detection of lysozyme in human urine with gold nanoparticles” by Castillo et al. deals with the possibility of easy and fast detection of urinary lysozyme achieved by colorimetric approach, employing gold nanoparticles as colorimetric agent.

The paper is enjoyable and easy to read, however not convincing.

It is from long time that colorimetric tests have been used in determining presence of particular chemicals, also with the purpose of detecting possible infections or deseases directly in human fluids.

Ten years ago Wang, X. et al. in “Direct determination of urinary lysozyme using surface plasmon resonance light-scattering of gold nanoparticles”, doi: 10.1016/j.talanta.2010.05.034 (which is by the way cited twice, number 6 and 12 in the reference list) have presented the application of gold NP as colorimetric probe in detecting lysozyme taking advantage of the aggregation effect that the protein has on AuNP. The only difference with respect to the present work is that they didn’t use urine.

In 2019, Santopolo et al. in “Ultrafast and Ultrasensitive Naked-Eye Detection of Urease-Positive Bacteria with Plasmonic Nanosensors”, doi: 10.1021/acssensors.9b00063, employed exactly the same approach for detecting bacteria in urine. In that case, the interaction mechanisms were clarified and all the experimetal conditions well defined.  

In conclusion, I do not find any reason to publish this paper as communication in Nanomaterials, first because it is not a novelty, second because the experimental procedure is really poor.